# Viral escape-inspired framework for structure-guided dual bait protein biosensor design

Yee Chuen Teoh[1], Mohammed Sakib Noor[2], Sina Aghakhani[3], Jack Girton[2], Guiping Hu[3], Ratul Chowdhury[2,4]*

**1** Department of Computer Science, Iowa State University, Ames, Iowa, United States of America, **2** Department of Chemical and Biological Engineering, Iowa State University, Ames, Iowa, United States of America, **3** School of Industrial Engineering and Management, Oklahoma State University, Stillwater, Oklahoma, United States of America, **4** Nanovaccine Institute, Iowa State University, Ames, Iowa, United States of America

☯ These authors contributed equally to this work.
* ratul@iastate.edu

## Abstract

A generalizable computational platform, CTRL-V (Computational TRacking of Likely Variants), is introduced to design selective binding (dual bait) biosensor proteins. The iteratively evolving receptor binding domain (RBD) of SARS-CoV-2 spike protein has been construed as a model dual bait biosensor which has iteratively evolved to distinguish and selectively bind to human entry receptors and avoid binding neutralizing antibodies. Spike RBD prioritizes mutations that reduce antibody binding while enhancing/ retaining binding with the ACE2 receptor. CTRL-V's through iterative design cycles was shown to pinpoint 20% (of the 39) reported SARS-CoV-2 point mutations across 30 circulating, infective strains as responsible for immune escape from commercial antibody LY-CoV1404. CTRL-V successfully identifies ~70% (five out of seven) single point mutations (371F, 373P, 440K, 445H, 456L) in the latest circulating KP.2 variant and offers detailed structural insights to the escape mechanism. While other data-driven viral escape variant predictor tools have shown promise in predicting potential future viral variants, they require massive amounts of data to bypass the need for physics of explicit biochemical interactions. Consequently, they cannot be generalized for other protein design applications. The publicly availably viral escape data was leveraged as *in vivo* anchors to streamline a computational workflow that can be generalized for dual bait biosensor design tasks as exemplified by identifying key mutational loci in Raf kinase that enables it to selectively bind Ras and Rap1a GTP. We demonstrate three versions of CTRL-V which use a combination of integer optimization, stochastic sampling by PyRosetta, and deep learning-based ProteinMPNN for structure-guided biosensor design.

**Data availability statement:** Code can be found at https://github.com/ChowdhuryRatul/CTRL-V. All other relevant data are in the manuscript and its supporting information files.

**Funding:** This work is supported in parts by the Iowa State University Startup Grant (Building a World of Difference Faculty Fellow), Center for Industrial Research and Service (CIRAS) Mini Grant, and NSF 22-599, Established Program to Stimulate Competitive Research (EPSCoR) RII Track-1, Award Number DQDBM7FGJPC5 to RC. MSN was funded by the NSF 22-599, Established Program to Stimulate Competitive Research (EPSCoR) RII Track-1, Award Number DQDBM7FGJPC5 grant. The funders had no role in study design, data collection and analysis, decision to publish, or preparation of the manuscript.

**Competing interests:** The authors have declared that no competing interests exist.

## Author summary

We developed CTRL-V, a computational platform that helps design dual bait biosensor proteins capable of selective binding between two proteins. Our work was inspired by the observation that rapidly pathogens like SARS-CoV-2 have transmembrane spike proteins which *sense* host proteins and evolve to minimize antibody neutralization while maximizing their ability to infect cells through entry receptors. We treated this viral protein evolution in nature as a model for designing dual bait biosensors. Leveraging viral evolution data, and a bespoke workflow, we tested and proved CTRL-V's ability to blindly predict mutations to the wild type spike protein that showed up in 30 infective, circulating SARS-CoV-2 strains. We recover 70% of the mutations in the latest KP.2 variant. CTRL-V generalizes to non-viral protein interaction systems as well as exemplified with Raf-Ras-Rap1a signaling pathway. This highlights the platform's versatility for designing dual bait proteins with selective binding properties. This broader applicability means our platform can be adapted to design proteins that discriminate between various molecules, potentially leading to better biosensors for environmental monitoring, healthcare diagnostics, and other applications. By providing detailed structural insights into why specific mutations affect binding, CTRL-V represents an interpretable approach for developing highly selective biosensors with diverse applications.

## Introduction

There is a significant industrial demand for selective biosensors in fields like environmental monitoring [1], healthcare diagnostics [2], food safety [3], and green energy production [3]. Specifically, detecting specific (heavy metal, toxin) contaminants in water or identifying disease biomarkers requires biosensors [4] with high specificity and sensitivity [5]. Rapid design is crucial to meet these needs, especially in responding to emerging challenges. Casting the biosensor design process as an optimization problem—with dual objectives of (a) enhancing binding to a target molecule, while (b) reducing or maintaining binding with another—enables the swift generation of such tailored protein libraries. These libraries can then be experimentally tested, accelerating the development of highly effective *dual bait* biosensors. The dual bait biosensor terminology has been introduced two decades back [6]. In this article we leverage the structural and biochemical changes to the key antigenic protein of SARS-CoV-2 (spike receptor binding domain; RBD) which makes it selectively recognize and bind to the human ACE2 receptor protein over human immune protein(s) (antibodies) [7], across several infective strains as an inspiration to construct a novel computational dual bait biosensor design platform.

Lack of technical readiness and detailed structural, biochemical, and mechanistic understanding for rapidly expanding SARS-CoV-2 variants has impacted every facet of human experience in the last few years, and the world is still recovering

from the loss (Fig 1). Globally, as of now, SARS-CoV-2 has infected over 700 million individuals, and the death toll has reached 7 million, while in the USA, a total of 1.2 million lives have been lost to the pandemic [8]. While the health sector was collapsing, the countrywide lockdown caused economies to falter, with the USA experiencing its highest unemployment rate since 1930, peaking at 4.9%. One in six renters could not catch up with their rent payments, while nearly one

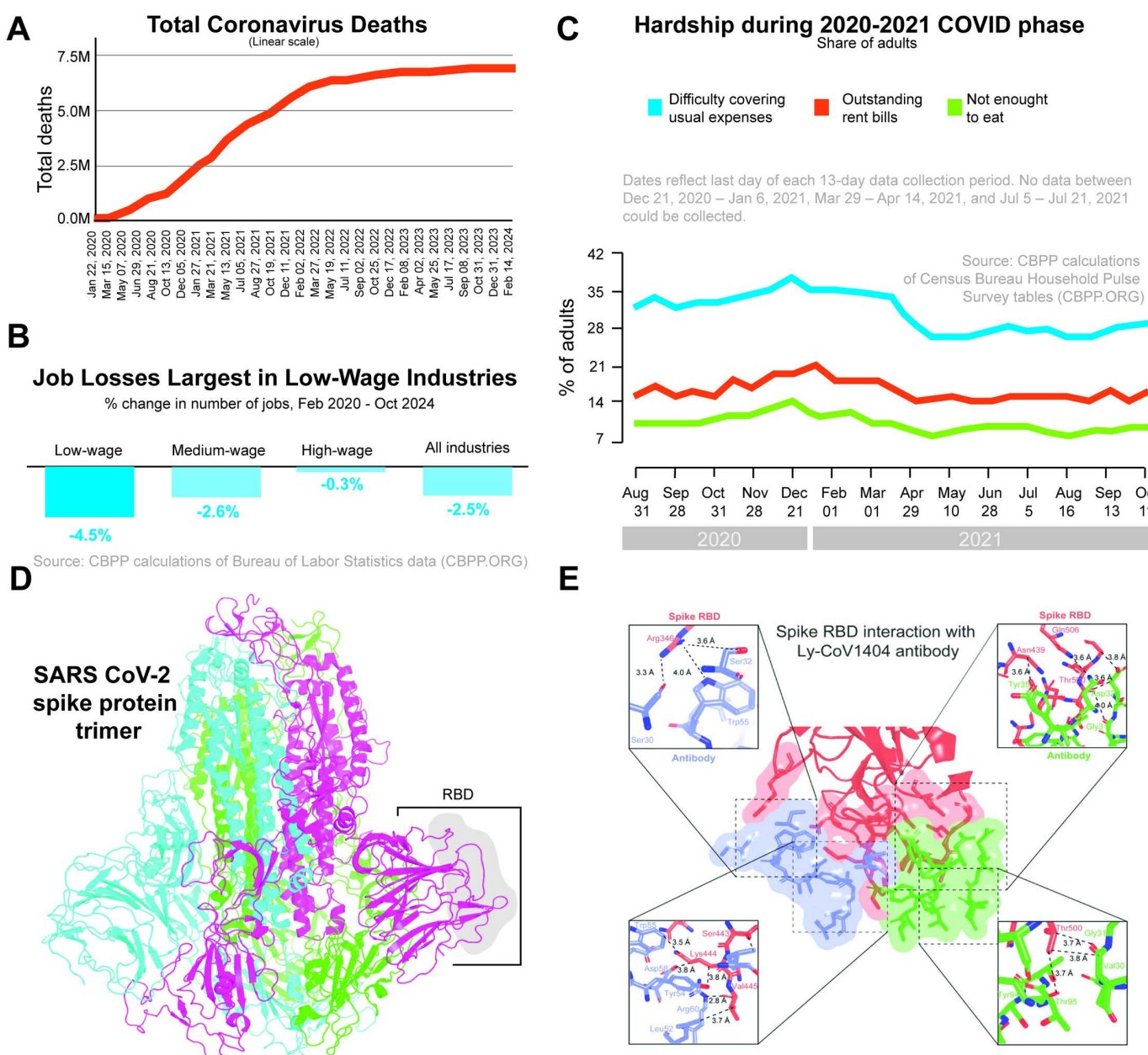

**Fig 1. The impact of SARS-CoV-2 (A-C) Total coronavirus deaths, loss of jobs, and percentage of adults facing financial hardships at a personal level.** This data was obtained from the Center on Budget and Policy Priorities (cbpp.org). (D) The overall structure of the trimeric spike protein of SARS-CoV-2 with the receptor binding domain (RBD) highlighted inset. (E) Interaction interface between RBD with commercial antibody LY-CoV1404. Insets illustrate the interacting residues from the RBD and heavy (blue) and light (green) chains of the antibody proteins within 4 Å.

in eight adults living in households with children reported insufficient food at the tail-end of 2021 [9]. At the onset of the pandemic, SARS-CoV-2 was gaining approximately two mutations a month in the global population [10], and since then, the World Health Organization (WHO) has recognized 11 critical variants of SARS-CoV-2 [11,12]. WHO has designated these variants under broad categories, such as variant of interest (VOI), variant of concern (VOC), and variant under monitoring (VUM) based on the transmissibility, severity, and immune response. Out of three, VOC has the highest transmissibility, increased severity, and increased immune evasion [13]. A study by Campbell *et. al* based on a 2021 worldwide data pool showed a significant increase in the pooled mean effective reproduction or transmission relative to non-VOC/ VOI of Alpha (B.1.1.7) at 29%, Beta (B.1.351) at 25%, Gamma (P.1) at 38% and Delta (B.1.617.2) at 97% [14]. A competitive comparison of transmissivity between these variants shows that Alpha and Beta had nearly identical transmissivity (differing by only ~4%), while Gamma exhibited an increase of 10% and 17% relative to Alpha and Beta, respectively [14]. Compared to Delta, Omicron, first identified in Botswana and South Africa in late 2021, was found to have a 5.4-fold weekly increase in cases thanks to overwhelming 30 mutations in its spike protein [15]. Andrews et al. demonstrated that two doses of primary immunization of ChAdOx1 nCoV-19 (AstraZeneca) vaccine were ineffective while that of BNT162b2 (Pfizer–BioNTech) vaccine observed a drop in effectiveness from 65.5% to 8.8% when tested against Omicron variant [16]. However, data suggests its clinical impact is less severe than its predecessors [17]. An epidemiologically inspired computational, structural biology platform technology (CTRL-V) to anticipate the clade of prospective infective variants of any rapidly mutating virus can provide a crucial biotechnological edge in combating global health challenges by informing the design of broadly neutralizing future proof therapeutic proteins. CTRL-V, by construction, allows for generalization beyond protein-protein discrimination tasks and can be readily adapted to design dual bait proteins ligands for metals, nucleotides, small molecules, and their binary combinations. Along the therapeutic front, CTRL-V predictions of future viral escape variant clade can expedite the development and biomanufacturing of rapid cross-neutralizing antibodies that provide protection against future variants, thereby equipping us with the tools necessary for prompt and robust responses to emerging threats.

### Viral infection

Respiratory viruses enter the human/ host through various molecular interaction routes at the epithelial lining of the trachea or lungs (bronchi). Viral capsid/ envelope proteins (spikes) bind to protein (such as angiotensin-converting enzyme-2- ACE2 for SARS-CoV-2) or carbohydrate-based ciliary receptors (such as sialic acid for influenza-A) on host epithelia. This binding event is the first step necessary for infection and facilitates the downstream deployment of the virus's genetic material (DNA or RNA) into the cell and subsequent utilization of the host cellular machinery for replication and dissemination to other parts of the body by entering the bloodstream [18].

The immune system initiates a response upon detecting the presence of foreign antigenic viral proteins in an attempt to neutralize them. This involves the release of T cells and B cells, both capable of recognizing and reacting to known antigenic epitopes [19]. T cells are responsible for eliminating infected cells, while B cells are responsible for generating antibodies. These antibodies possess the capacity to bind to antigens within the body by attaching their paratopes to the antigenic epitopes [20]. This binding event between the antibody and antigen facilitates the recognition of antigens by phagocytotic cells, ultimately leading to their elimination [20]. However, for novel viruses, the body fails to provide humoral resistance, and inoculation of extraneous antibodies through vaccination becomes necessary to combat the viral infection.

For example, SARS-CoV-2 spike protein mediates virus attachment to host cell-surface receptors (ACE2) and fusion between virus and cell membranes [21]. This transmembrane glycoprotein consists of two subunits, S1 and S2. S1 contains the amino-terminal domain and the receptor binding domain (RBD; **Fig 1**) and S2 includes the trimeric core of the protein and is responsible for membrane fusion[22]. RBD is a highly variable region as predominant mutations occur in this domain, which allows the virus to reduce the neutralizing efficacy of the antibodies and enhance the binding with the ACE2 [22,23]. Viral escape, thus, refers to mutation events in the RBD that allow for the evasion of antibodies and

increased ACE2 binding. For instance, the mutation N439K first emerged in Scotland in March 2020, which was reported to result in high binding affinity with the ACE2 receptor but decreased the neutralizing effect of the sera antibodies [24]. In the scope of the study, we will be looking into the landscape of single-point mutations in the RBD based on simultaneous reduction in affinity towards a commercial antibody while maintaining/ improving affinity towards the ACE2 receptor (i.e., entry protein).

## Capturing Virus Mutation as a proof-of-concept inspiration for dual bait biosensor design

Traditional vaccine response to viral outbreaks is effective but to a degree and not totally future proof. *A priori* knowledge of viral mutations that rise to more infective and more lethal variants requires understanding underlying principles and constraints that guide viral evolution. Various computational methods have been explored to different extents in predicting viral mutation to understand the fundamental tenets of viral mutation preferences and latest protein structure prediction methods (AlphaFold2 [25], OmegaFold [26], RGN2 [27]) have alluded to them (albeit with little success). Recent advances in computational structural biology have led to various approaches for predicting protein-protein interactions and viral mutations, yet each address only partial aspects of the complex challenge of viral escape prediction. Machine learning approaches have led recent developments, with notable advances such as the language model-based approach by Hie et al [28] innovatively applies natural language processing to viral evolution, achieving over 85% prediction accuracy for SARS-CoV-2 mutations by treating viral sequences as linguistic constructs. Similarly, EVEscape [29] combines deep learning on historical viral sequences with biophysical information to predict mutation fitness effects. While these approaches excel at capturing evolutionary patterns, they fundamentally rely on massive corpus of sequence data, potentially limiting their any interpretability on the physics of binding and infectivity. Consequently, they cannot be applied more broadly as a protein engineering platform to design dual bait biosensors that discriminate between different interaction partners. Current tools thus implicitly optimize a latent sequence fitness objective for viral mutation prediction and cannot provide any structural underpinning of how mutations help to discriminate between different interaction partners. A study by Salama *et al.* proposed a novel machine-learning technique for mutation prediction that appears on primary RNA sequence-structure alignments [30]. This study predicts the genotype of each nucleotide in the RNA sequence and proves that a nucleotide in an RNA sequence change based on the other nucleotides in the sequence [30–32]. Sequence-based interaction tools like MaSIF [31] and DeepPPISP [32] focus specifically on protein-protein interaction prediction. MaSIF employs geometric deep learning for surface pattern recognition, while DeepPPISP predicts interaction sites using sequence information alone. However, these tools are not trained on viral proteins given the low number of experimentally crystallized structures exists for both (a) viral protein and host receptor, and (b)) viral protein and neutralizing antibody. This is why such tools are poised/ trained to capture the dynamic nature of viral evolution or the simultaneous interaction with multiple proteins.

More specialized tools have recently emerged for antibody-antigen interactions. mCSM-AB2 [33] uses graph-based machine learning to predict antibody-antigen binding affinity changes, but it ignores the interaction between antigen and receptor and has (a) an incomplete biochemical objective function, and hence (b) cannot be applied for dual bait biosensor design involving three proteins and two paiwise interactions.

PRIEST [34], much like EVEscape, employs deep learning on massive corpuses of temporal sequence data to predict mutations which focuses solely on evolutionary patterns of the sequence with no awareness of structural biochemistry, or direct selection pressure imposed by specific host antibodies and host receptor proteins.

A multi-task learning approach by Gross and Sharan et al [35] represents a step forward by analyzing experimental escape data from multiple antibodies simultaneously but remains specifically focused on antibody escape in SARS-CoV-2's receptor binding domain without addressing (a) receptor binding, (b) lack of structural data, and (c) energetics of binding. Other works on the other hand focus on the antigen-receptor interactions [7,36,37] and are completely unaware of antibody information.

CTRL-V to that end, provides structural insight to individual residue-level contribution of energetics and a strong biochemical prior for understanding why and how proteins of interest can discriminate between different interacting proteins (and potentially, metal ions, nucleotides, and small molecules). CTRL-V distinguishes itself from these existing approaches by (a) structural characterization of selection pressure on viruses, and (b) unlike tools dependent on historical data or experimental training sets, CTRL-V employs a modular, physics-based framework that integrates established computational tools to analyze protein-protein interactions without requiring prior training data.

We demonstrate three different computational frameworks (within CTRL-V) geared to the same biochemical objective of taking as input the SARS-CoV-2 spike RDB, human angiotensin converting enzyme 2 (ACE2) receptor, and commercial neutralizing antibody LY-CoV1404 protein structures, and predicting as output point mutations to RBD that enables the greatest differential binding between the other two proteins. All CTRL-V versions follow the same modular system design architecture as shown in **Fig 2**. It allows modular plug-and-play of state-of-the-art machine learning-based, force-field based, and a novel *dual objective* integer optimization program [38] (; similar work on protein design [39]) which predicts RBD mutations which allow for non-compromised interaction with entry human ACE2 receptor and reduced interaction with when confronted with a known neutralizing antibody (LY-CoV1404). This allows CTRL-V to not only unveil a putative viral mutational landscape but also infer the right combination of existing and in-house tools that must be used for an analogous dual bait biosensor design, with least uncertainties. This model system was chosen since experimentally resolved crystal coordinates of RBD-ACE2, RBD-LyCoV1404, and mutational data across 30 circulating variants that arose from the wild type RBD with some contribution from the globally administered LyCoV1404 antibody.

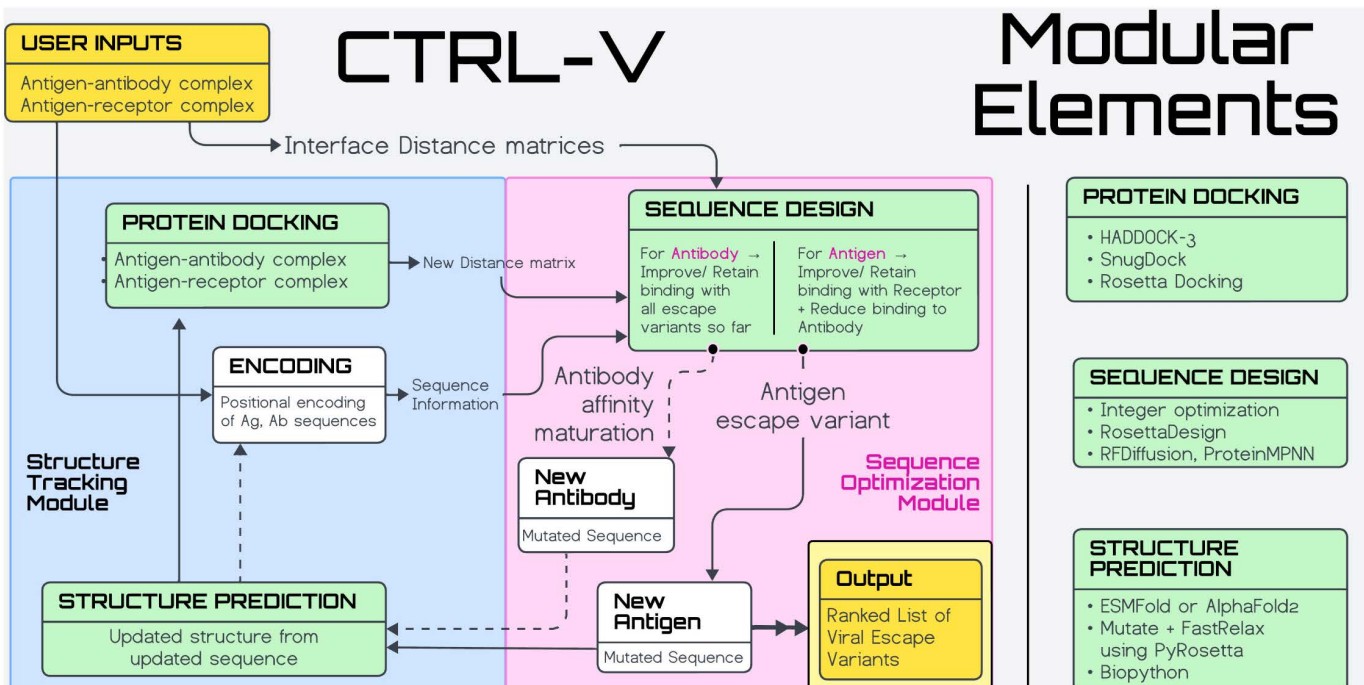

**Fig 2. CTRL-V workflow architecture.** The modular elements (green) are included in the iterative loop. The blue box represents the structure tracking module where updated antigen (and optionally antibody) structures are predicted, and antigen-antibody docking is performed. The red box represents the sequence optimization module where the new sequences (for antigen and antibody) are predicted and evaluated. The inputs and outputs are marked in yellow.

Identifying a clade of future variants likely to escape human immune neutralization is critical information to develop long-lasting, antiviral therapies and add to global biotechnological readiness in allaying pandemics. It also provides a structural basis to rapidly, computationally assess viral escape potential from *de novo* designed antibodies. Our work focuses on the viral mutations that are seen in the receptor binding domain (RBD) of viral spike protein, the predominant mutational locus, responsible for the evasion of the immune clutch. While some mutations do not bring about a change in the infectivity or severity, others make the case for detailed investigation. For a dual bait biosensor design using CTRL-V, the design-protein (DP), intended target molecule (TM), and non-target molecule (TM*) will replace the viral spike, the human ACE2, and the Ly-CoV1414 antibody proteins, respectively.

## Results

### Use of experimental data from SARS-CoV-2 for benchmarking CTRL-V fidelity

In evaluating the performance across three versions of CTRL-V, we use experimentally confirmed crystallographic coordinates of commercially available antibodies (LY-CoV1404 and its single-chain nanobodies) bound to SARS-CoV-2 spike RBD (PDB accession id: 6M0J [40,41]). LY-CoV1404, also known as Bebtelovimab [42,43], is a monoclonal antibody that receive emergency use authorization for treatment of COVID-19 from February 2022 to November 2022 [44–46]. We ran CTRL-V simulation trajectories in parallel for the antigen-antibody (heavy, light, and both chains separately) complexes of LY-CoV1404, 7Y0W RA, 7Y0W RB, 7Y0W RH, and 7Y0W RL. Results from all these simulations were collated and analyzed.

To evaluate the performance of the approaches and versions, we identified the 30 most popular variants circulating in the last three years across the globe (as reported by Covariants [11]; see S1 File). Among the 30 variants, there are 39 different single-point mutations, as shown in Fig 3, which shows that the virus carefully screens through an exhaustive sequence of landscape mutations and selects only a few specific ones that satisfy the biochemical/ physical objective. The objective can be, for example, to escape the antibody, survive in a specific environment, cell division, and much more. While all mutations seen across circulating variants are not a result of viral escape, we wanted to use this study to explore how many were. We permitted mutations to a 195 aa non-contiguous, solvent-exposed stretch on the viral spike RBD. Consequently, there were $20^{195}$ possible mutations exhaustively, and the combinatoric explosion therein makes it computationally intractable, even with the most powerful computing architectures today. Hence, instead, we look at all possible single-point mutations, a total of 3900, with the objective to biophysically rationalize (reported in **Table 1**) which of these mutations SARS-CoV-2 are instrumental for immune escape. The predicted frequency of each mutation in the simulated viral escape trajectories, and computational binding scores with human ACE2 were corroborated against experimental deep mutational scanning data on these 3900 point mutations for ease of expression and binding affinity with human ACE2 receptor. While a classical definition of recovery (computationally) refers to the fraction of truth set correctly identified by a computational protocol, here, our truth set is the list of truly escape variants of SARS-CoV-2. Since it is so far unknown, experimentally for all 39 point mutations, whether their biochemical objective is solely antibody escape (that too from this specific LY-CoV1404 antibody), we have used recovery rate as a computational metric to categorize how many of these point mutations are likely to be escape mutations for the specific antibody only. This then allows us to list putative, future escape mutations as per our Favorable Mutation Problem (FMP) criterion which have so far not been seen/ reported in widely circulating infective strains but could be emergent in upcoming variants. This predicted set of putative escape mutants is useful input for downstream antibody design studies like prior efforts [47–51].

### Accuracy, precision, recall, and biological significance of these numbers

All three CTRL-V approaches (or versions; used interchangeably) are evaluated starting at the viral protein sequence in the antigen-antibody complex protein (**Table 1**). All approaches are assessed on their accuracy, precision, and recall

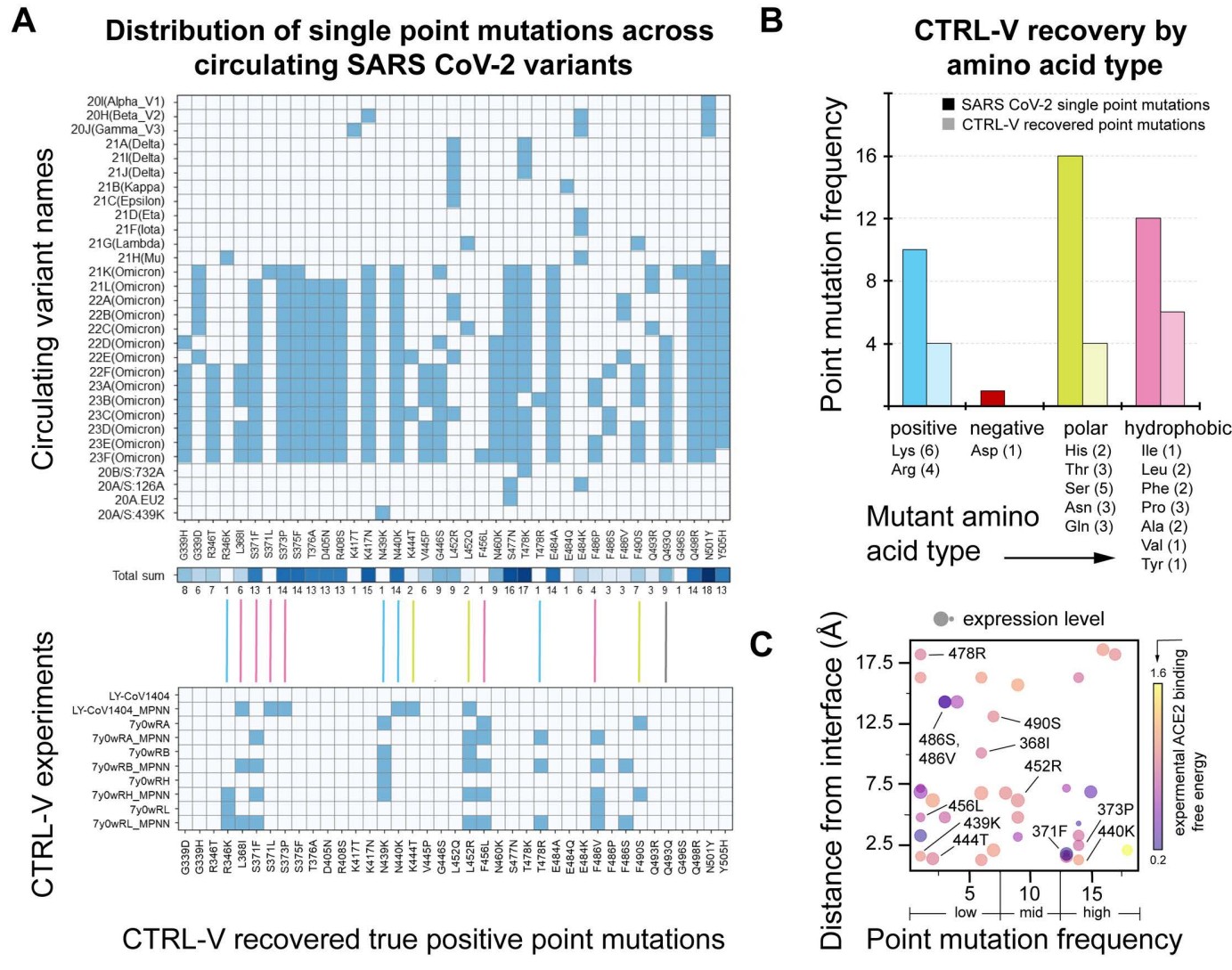

**Fig 3. Details about circulating SARS-CoV-2 variants.** (A) SARS-CoV-2 mutations and their frequency of appearance, and chemical types of point mutations in all circulating variants compared to prediction from the CTRL-V simulations. Blue represents positive, red negative, green polar, and pink hydrophobic amino acid transitions to the spike proteins. The bottom panel shows those point mutations that were correctly identified across different CTRL-V simulations. (B) Illustrative insight into the favored mutations in circulating infective variants across their amino acid chemical categories shows CTRL-V can successfully replicate the variety of point mutations: positive, polar, and hydrophobic. (C) Distribution of how far each mutation is from the antigen-antibody binding interface, its frequency across circulating variants, its binding affinity towards the ACE2 receptor, and how well it is expressed on the viral surface for each of the 39 mutations (from an experimental deep mutational scan study). CTRL-V-recovered mutations are labeled.

percentage. Due to the imbalance in true mutations and false mutations on SARS-CoV-2 single-point mutations with only 39 true mutations and 3861 false mutations (i.e., mutations that were not observed in circulating strains), the accuracy percentage has trivial significance as all three CTRL-V versions predict nearly all false mutations correctly and possess a 99% accuracy. Hence, we focus on the recall percentage, which tells us how many point mutations out of the 39 circulating point mutations can these versions predict. The primary utility of CTRL-V lies not in serving as a state-of-the-art escape mutation predictor, but as a biosensor design platform, refined across three versions using viral escape data to evaluate model quality.

**Table 1. Approach I: CTRL-V performance (accuracy, precision, and recall) using three different versions (version 1: with integer optimization, version 2: with stochastic sequence design and docking in PyRosetta, and version 3: with deep learning-based model ProteinMPNN).**

| Method | Performance metric | Antibodies used (separate simulations for each) | | | | |
|---|---|---|---|---|---|---|
| Version 1–Integer optimization | | LY-CoV1404 | 7Y0W_RA | 7Y0W_RB | 7Y0W_RH | 7Y0W_RL |
| | Accuracy | 98.56% | 98.01% | 98.78% | 98.76% | 97.85% |
| | Precision | 5.26% | 2.50% | 0.00% | 0.00% | 0.00% |
| | Recall | 2.56% | 2.56% | 0.00% | 0.00% | 0.00% |
| Version 2–PyRosetta | Accuracy | 98.46% | 97.67% | 97.41% | 97.49% | 97.41% |
| | Precision | 0.00% | 5.17% | 3.03% | 1.64% | 3.03% |
| | Recall | 0.00% | 7.69% | 5.13% | 2.56% | 5.13% |
| Version 3– Protein-MPNN | Accuracy | 98.51% | 98.08% | 98.05% | 98.36% | 98.05% |
| | Precision | 19.35% | 10.87% | 15.09% | 17.95% | 15.06% |
| | Recall | 15.38% | 12.82% | 20.51% | 17.95% | 20.51% |

Since there is (a) non-comprehensive determination on how many of these 39 mutations arose to escape against this specific LY-CoV1404, and (b) reported polymorphisms in human ACE2 from various ecotypes [52] (i.e., ACE2 sequences are seen to vary across geographical niches), we are using recovery rate as a computational metric to evaluate CTRL-V model quality and utility as a biosensor design platform. CTRL-V version 1 (CTRL-V-1) unfolds that the quality of the underlying pairwise amino acid interaction scores (both at the DP-TM and DP-TM*interfaces) dictate the performance of the integer optimization model. We progressively finetune the choice of appropriate tools (in the latter versions) to segue into a powerful predictive model (**Table 1**). While version 3 (CTRL-V-3) is the most performative model, we still wanted to discuss what we gleaned from the shortcomings of versions 1 and 2 and why each specific modification was made to arrive at CTRL-V-3. This is an attempt towards putting out negative results and negative simulations for the computational biology community to be aware of why our final version worked better than others and also to provide a recipe to improve version 1 for future endeavors (as it remains most effective computationally).

## Investigation and explanation of CTRL-V version 1 performance

CTRL-V-1 uses an integer program to predict point mutations to the antigen (and optionally to the antibody) reliant on an experiment-derived scoring function and state-of-the-art tools for the remaining steps: (a) for structure prediction and updating DM structures, and (b) HADDOCK3 for flexible docking between DM-TM and DM-TM*. We notice an improvement in the prediction quality in recall percentage after switching state-of-the-art tools in CTRL-V-1 to more conventional PyRosetta in CTRL-V-2 for both updating structures and docking. From the RMSD data of point mutants CTRL-V-1 (**Fig 4**), it is observed that the structures rapidly worsen (unfold) with every single mutation in ESMFold. When investigating the structural predictions by ESMfold, a significant deviation (at least 24Å) in the viral protein structure is observed even with one point mutation (**Fig 4**). This is expected, given the poor homology across viral proteins and the low frequency of such structures in the training sets [53]. Subsequently, HADDOCK3 reasoned over these worse, wrongly folded antigens (DM) and antibody (TM*) structures and led to unreasonable results (such as splitting of heavy and light antibody chains; **Fig 4**), thereby providing the integer optimization model with inaccurate interface distance information. Furthermore, the amino acid preference score reported by Jha *et al* [54]. (**S1 Fig**), albeit experimentally derived, is utilized to calculate the mutation preferences of both the viral and antibody proteins at each sequence position and to simulate interactions and mutation preferences in the binding regions. Because the scoring system is empirically derived from a broad class of proteins as opposed protein-protein interfaces, it ended up biasing the integer optimization heavily towards favoring prolines and disfavoring cysteines. The preference score table depict heavy repulsion between prolines and strong attraction between

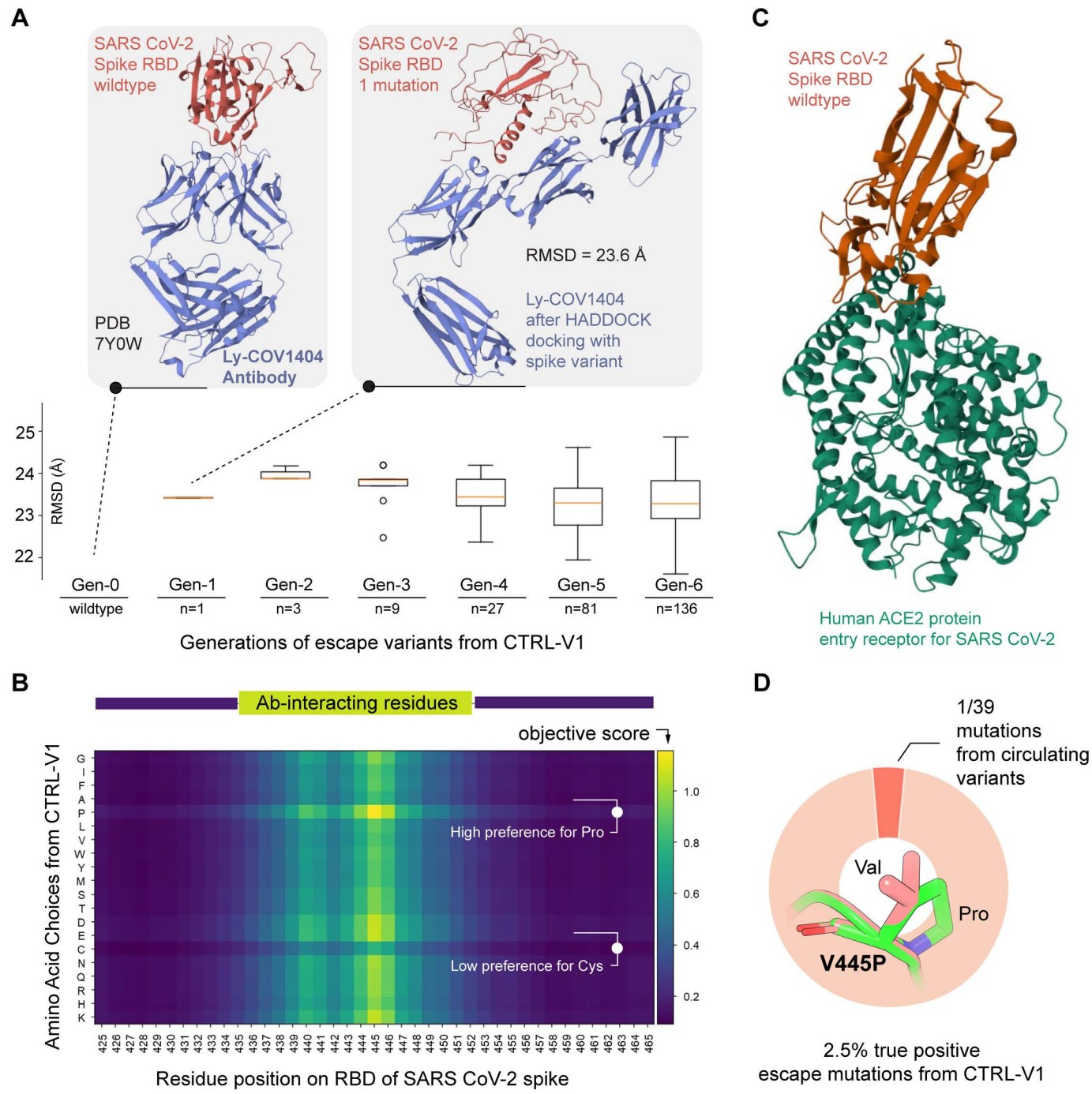

**Fig 4. Performance of CTRL-V version 1.** (A) ESMfold predicts mutated viral spike RBD shows worsening of prediction quality due to paucity of viral proteins in training data. The RMSD plot shows that worse structures get carried through to newer generations and never get fixed (i.e., maintain ~24Å RMSD with starting structure). (B) The maximized objective function in the integer optimization model on all possible single-point mutations for viral protein at the interface. There are two distinct horizontal swaths (indicated with white text): a high preference for proline and a low preference for cysteine. (C) Interaction of wildtype SARS-CoV-2 spike RBD with its human entry receptor protein ACE2. (D) CTRL-V version 1 captures only one true, circulating single-point mutation, V445P, out of 39 known point mutations (likely because version 1 generates a high number of proline variants).

cysteines and all other nineteen amino acids. Given the biochemical objective is to eliminate DM (viral spike RBD) binding with TM* (antibody) and improve/ maintain binding with TM (human ACE2 receptor), the DM accumulates all proline mutations - which is un-realistic. As shown in **Fig 4** (details, S2 Fig), a distinct horizontal patch indicates a high preference for proline and a low preference for cysteine. This is because the scores reflect that most amino acids are biochemically repelled by proline, much stronger than the strongest attractive forces, even between salt-bridging amino acids. We surmise that finetuning these pairwise amino acid interaction scores to account for pKa values of side chains, their hydrophobicity indices (as per Kyte and Doolittle scale [55]), scaled by 3D-distance of separation between two interacting amino acids, would be a reasonable, but separate endeavor. Hence, we chose to limit the usage of this biased scoring table to only CTRL-V-1, as a baseline to demonstrate how other scoring methods lead to better performance, albeit at higher computing costs. We discussed CTRL-V-1 as an archetype for demonstrating how model performance and investigation can inform process flow optimization. This is an attempt to illuminate the process of workflow optimization which is hard to find in current studies which are inundated with opaque, machine learning models which in most cases do not generalize well and offer little-to-no interpretability to domain experts.

## PyRosetta and rigid body docking to resolve scoring and docking multichain target molecules

As mentioned, CTRL-V-1 an integer optimization formulation to predict the favorable amino acid mutation on the DM (spike RBD), ESMFold to predict the DM variant structure, and HADDOCK to perform DM-TM* (antigen-antibody) and DM-TM (antigen-receptor) docking. Reliance on the empirical amino acid interaction scores in the optimization step leads CTRL-V-1's DM mutations to be heavily biased towards proline (from Jha *et al* [54]). Also, ESMfold was unable predict structures of DM variants (as DM is a viral protein), which affected downstream HADDOCK docking. To this end we built CTRL-V-2 with PyRosetta with rigid body docking to resolve such bias. PyRosetta's *InterfaceAnalyzerMover* (energy function) for binding energy calculations and *mutate_residue* module was used to construct the DM variant structures and ranking them in CTRL-V-2. A breadth-first search explores all 20 amino acid choices at each of the fifteen polypeptide loci of the spike protein RBD (antigen; DM). Aside from that, with the inclusion of PyRosetta, we see that the variants, even after relaxation and energy minimization, have very similar antigen structures (RMSD <0.5Å) for most (>93%) of the produced variants, corroborating with known experimental observation [56].

## Inclusion of ProteinMPNN to explore combinatoric sequence design

To assess the effect of all possible $20^{195}$ single point mutations in the 195 aa residue-long RBD is computationally prohibitive for a brute force method. The addition of ProteinMPNN (Protein Message Passing Neural Network) allows for identifying all point mutation combinations to the DM that conserve the structure thereby unraveling the viable sequence landscape for DM. As we inspect the predictions from CTRL-V-2 with PyRosetta in comparison with CTRL-V-3 with ProteinMPNN, the latter was able to immediately eliminate single-point mutations that seem ideal at escaping antibodies (unbinding TM*) while binding to receptor (TM) but makes the spike RBD (DM) structure unstable. For example, PyRosetta (in CTRL-V-2) ranks single-point mutations 498I, 501P, and 501V as the top 3 favorable mutations to wild type RBD, while ProteinMPNN (CTRL-V-3) eliminates those mutations and selects 501A, 501S, and 444T single-point mutations. Consequently, CTRL-V-3 shows the successful recovery of epidemiologically reported, known SARS-CoV-2 infective mutation, 444T, and marks it as responsible for viral escape, which is experimentally corroborated [57]. CTRL-V-3 thus emerges as a viable dual bait biosensor design platform. We validate this generalizability using a non-viral example (Raf kinase-Ras-Rap1a signaling pathway) where CTRL-V-3 can identify key mutational locus on Raf kinase which help it selectively bind to Ras and not Rap1a GTPase enzyme.

## Biological significance of recovery in context with experimental deep mutational scan data

Looking at the performance of across three versions, CTRL-V-3 with ProteinMPNN performed best successfully categorizing 20% of point mutants from circulating variants as important for escape against Ly-CoV1404 antibody. While 20% does not seem like a high number when it comes to the recovery of true positives by a computational algorithm, the case is more biologically nuanced here. Mutations in viruses can be ascribed to a multitude of biological factors such as survivability, cell division, environmental adaptation, and much more. Since each point mutation in a circulating variant is not necessarily indexed as solely due to escape, that too from a single antibody provided in the vaccine (in isolation from the overall innate humoral antibody response), and a specific ACE2 ecotype. CTRL-V gives us a good indication that ~20% (8/39) of viral mutations facilitate viral escape against a specific commercial antibody. Again, the role of the circulating variant data is to provide an anchor to optimize the right workflow for a dual bait biosensor design platform. It has been optimized through three workflow iterations spanning >5000 design trajectories, validated on known *in vivo* data, and shown to work well on non-viral systems.

Fig 5 depicts the experimental log10 binding and expression scores from a deep mutational scanning of the SARS-CoV-2 receptor binding domain by Greaney *et al* [58] (see S2 File) and unravels insights about the characteristics of the

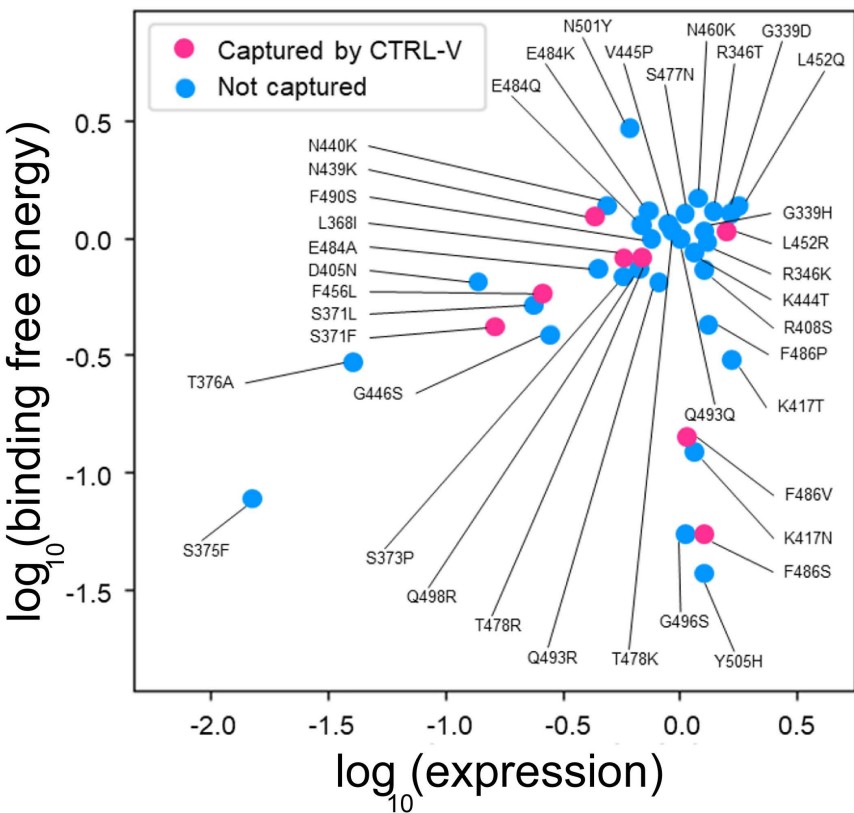

**Fig 5. Expression and binding score for all 39 SARS-CoV-2 single-point mutations.** Experimental data was reported from a deep mutational scan study by Greaney *et al.*[58]. Escape mutations predicted by different versions of CTRL-V are shown in pink. Neutral binding affinity and moderate expression score are more popular choices for circulating variants. CTRL-V captures the topology of the spread-out landscape by recovering at least one variant within 0.5 $\log_{10}$ binding free energy with human ACE2 receptor protein and 1 $\log_{10}$ expression of any known circulating variant.

39 infective point mutations in circulating variants. A high binding score indicates a higher binding affinity with ACE2, while a low binding score reflects weak binding. Variants with higher expression scores mean the virus expresses such variant spikes easily, or more copies of the variant per unit surface area on its surface, and lower expression scores indicate poor copy numbers. The data clustering of the 39 SARS-CoV-2 single-point mutations around the origin (**Fig 5**) indicates that neutral binding affinity and moderate expression score are more observed across these circulating variants. CTRL-V-3 indicates 8/39 (20%) single-point mutations to be truly escape variants (from the Ly-CoV1404 antibody) and demonstrates that the spread in their experimental expression-energy landscape is well captured by CTRL-V predictions.

### Ranking antibody binding score and ACE2 binding score

CTRL-V-3 predicted mutations to viral spike RBD (design molecule; DM) were ranked as top based on computational binding affinity (using PyRosetta energies) with Ly-CoV1404 antibody (non-target molecule; TM*) and ACE2 receptor (target molecule; TM) in ascending order of binding energies. A high binding score (free energy of interaction) indicates weaker binding between two proteins, and low scores indicate stronger binding. This was set up within CTRL-V-3 (mode I) to rank single-point mutations for antigen-antibody (DM-TM*) complex interaction in descending and rank single-point mutations for antigen-receptor complex (DM-TM) interaction in ascending. However, upon scoring the known 39 infective point mutations, it was clear that the virus does not always select amino acids with the highest binding energy against antibody and lowest binding energy against receptor (ACE2) as shown in **Fig 6**. So, we ran an additional variation of CTRL-V-3 (mode II), where we predict mutations that improve binding(ascending) to both ACE2 and the antibody. Such a scoring tactic (also explored before in enzyme engineering to identify designs that improve binding to a specific substrate and eliminate binding to another [59]) shows better agreement of energy trends and recovery in viral escape variants. This shows that stabilizing (i.e., better interacting) mutations are always helpful for the integrity of the designed protein as much as it is for an interaction.

### Structural biophysical underpinning and characterization of escape mutations in KP.2 variant

The SARS-CoV-2 variant KP.2, a descendant of JN.1, has emerged as a rapidly spreading lineage. Kaku *et al* [60] investigated the virological properties of KP.2. Compared to JN.1, KP.2 possesses three substitutions: two in the S protein and one in a non-S protein. Analyses based on genome surveillance data from the USA, UK, and Canada suggest that KP.2 has a higher effective reproduction number (Re) compared to JN.1 in these regions. This indicates a potential for KP.2 to become the dominant lineage globally. While infectivity assays showed that KP.2 is significantly less infectious than JN.1, neutralization assays revealed a different picture. Sera from individuals vaccinated with the XBB.1.5 vaccine and those with breakthrough infections from various SARS-CoV-2 variants exhibited significantly lower neutralization activity against KP.2 compared to JN.1. Notably, the most significant reduction in neutralization was observed in unvaccinated individuals who received the XBB.1.5 vaccine, suggesting that KP.2 has increased immune resistance. This increased immune escape likely contributes to the higher Re of KP.2.

CTRL-V-3 successfully identified five out of seven (371F, 373P, 440K, 445H, and 456L) mutations on the spike protein of KP.2 with causal biophysical underpinning that explains why these mutations enable the variant to escape the LY-CoV1404 antibody, used in COVID-19 commercial vaccine recipe. This also serves as a demonstration that CTRL-V is not only a promising biosensor design platform, but it also provides full structural interpretability for its predictions which enables it to parse experimental corroborations and false positives with certainty.

We hereby present the first biophysical characterization of the interaction between the spike receptor binding domain of KP.2 and the LY-CoV1404 antibody, providing a molecular explanation for the reduced neutralization efficacy of this antibody against this specific variant (**Fig 7**). These point mutations either introduce incompatible electrostatics or steric repulsion to reduce the LY-CoV1404 antibody's affinity for the spike protein, thereby enabling viral escape. Phe371 and Pro373 mutations introduce a very hydrophobic microenvironment at ~7Å distance from the Ser103 (chain A) and Asp56 (chain B)

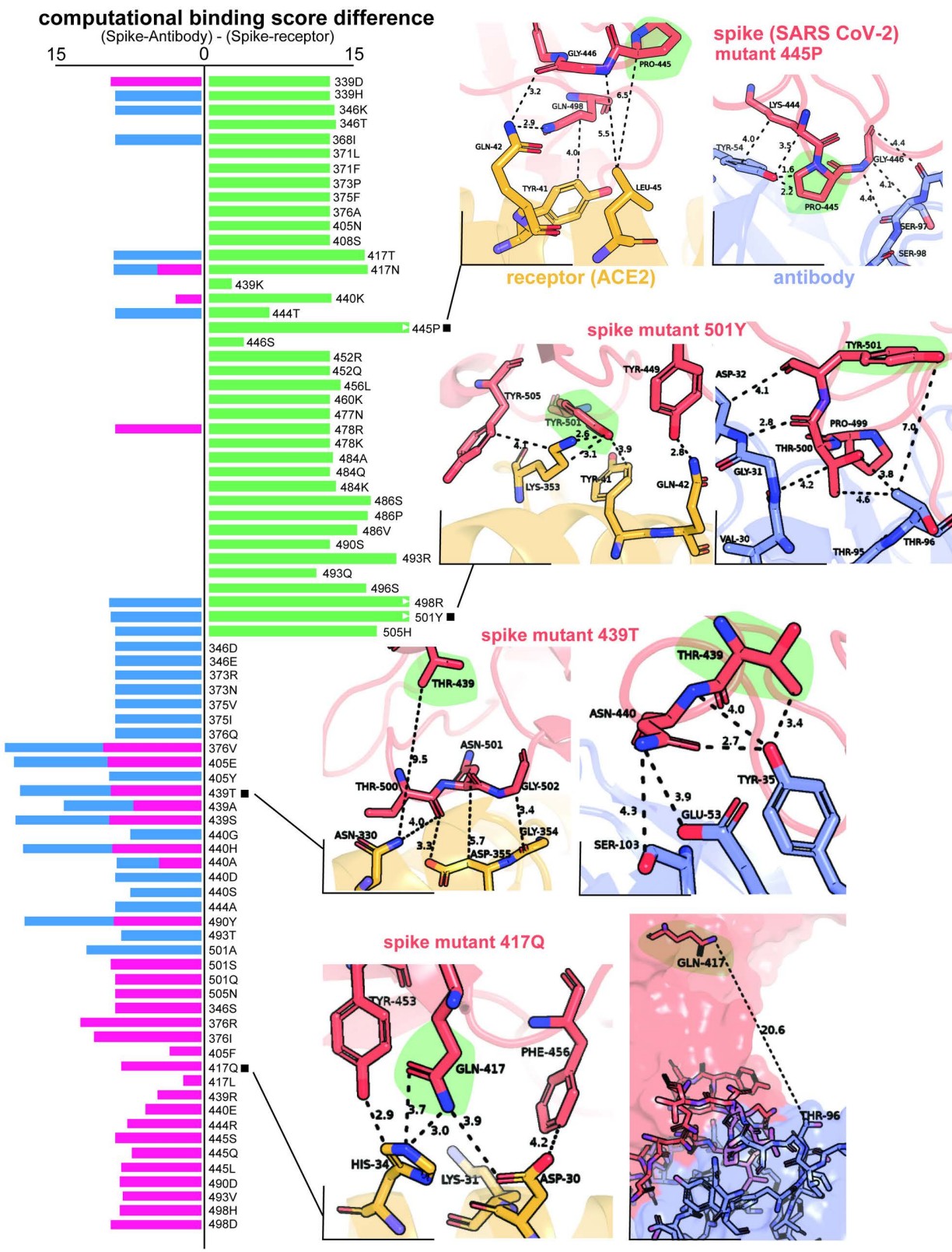

**Fig 6. Difference in binding energy between antigen-antibody complex and antigen-receptor complex.** Antigen refers to SARS-CoV-2 spike protein, antibody is LYCoV-1404 neutralizing antibody, and receptor is human ACE2 protein. The left bar plot lists all the mutations and corresponding

differences in the binding energy of the antigen-antibody complex over the antigen-receptor complex. Green stands for mutations in circulating variants, blue and red for (two differently posed objective functions for sequence design using) CTRL–V version 3. The first implementation (Mode I, red) improves binding to the receptor and lowers binding with the antibody and is destabilizing for the antigen itself, while the second implementation (Mode II, blue) identifies point mutations in the spike which are stabilizing to the spike and improve binding to both the receptor and antibody. Insets illustrate the key interacting residues at the interfaces of antigen (red) – receptor (yellow) and antigen (red) – antibody (blue) complexes.

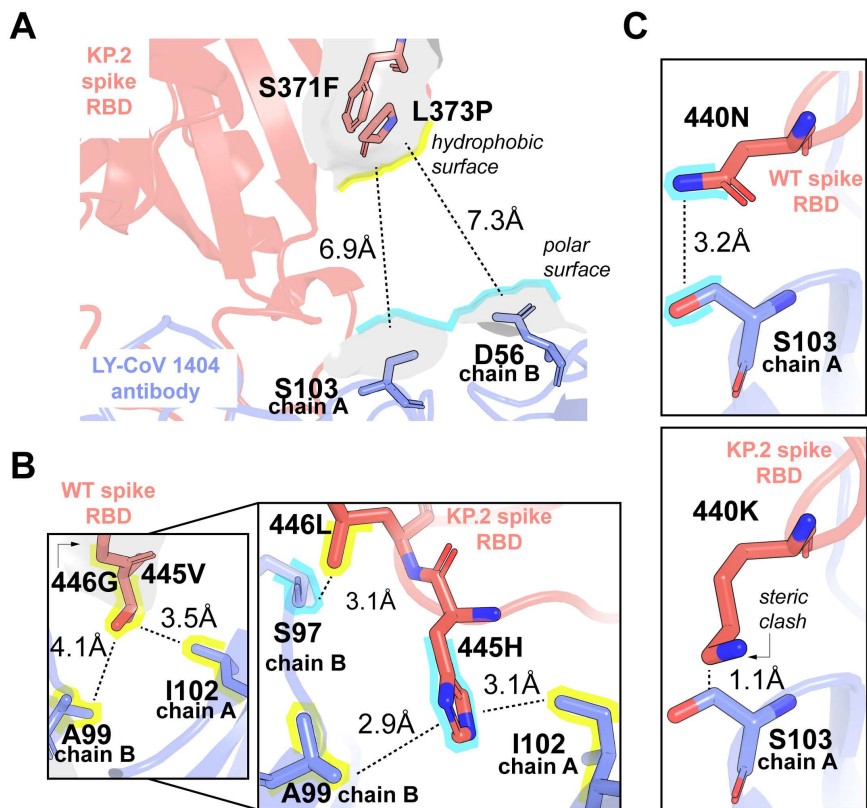

**Fig 7. Biophysical characterization of the KP.2 variant shows the five point mutations that CTRL-V deems to be causal for this variant to escape the immunity of LY-CoV1404 commercial antibody.** (A) S371F and L373P mutations on the spike introduce a hydrophobic patch close to a hydrophilic sub-surface of the antibody thereby allowing the spike to lower its affinity for the antibody and thereby escape. (B) V445H and G446L mutations together introduce a similar incompatible electrostatic surface through the insertion of polar groups close to hydrophobic ones and vice versa, respectively. (C) N440K mutation does not compromise the electrostatic microenvironment with the Ser103 side chain of the antibody but weakens interaction strength due to steric hindrance posed by the larger Lys440 side chain.

of the LY-CoV1404 antibody-spike interface. His445 disrupts the hydrophobic packing at the antibody-spike complex (originally mediated by Val445) by placing a polar side chain within ~3Å of a hydrophobic interface domain of the antibody (Ala99 from chain A and Ile102 from chain B). Similarly, the WT SARS-CoV-2 spike uses the backbone interaction of Gly446 for electrostatic attachment with the side chain of Ser97 (chain B of antibody). However, in the KP.2 spike, Leu446 mutation creates a hydrophobic microenvironment, thereby leading to loss of attachment with the polar Ser97 side chain. Finally, mutation of 440N to 440K in the KP.2 variant destroys the electrostatic contact between Asn side chain with Ser103 of antibody by steric clashes with a 1.2Å longer side chain conformation and 72% more bulk (80.8 and 139.1 Å$^2$ available surface areas for Asn and Lys, respectively). These mutations, recovered through CTRL-V simulations and structural information unfolds the molecular level insight about why KP.2 variant is reported to escape the administered LY-CoV1404 antibody.

## Cross checking with antibody Escape Estimator

The Escape Estimator by Greaney *et al.* is a computational tool designed to estimate the antigenic effect of arbitrary combinations of mutations in the viral protein RBD region [61]. Escape Estimator aggregates experimental data from deep mutational scanning studies [61] of 36 antibodies including 3 antibodies that were drawn out by SARS-CoV-1. Greaney *et al.* aggregates multiple antibodies, allowing researchers to assess how combinations of mutations may affect the virus's ability to escape binding from antibody, i.e., the antigenic effects of arbitrary combinations of mutations. To assess the fidelity of CTRL-V as an effective biosensor design platform and the suitability of the viral escape as a valid system to benchmark it we cross checked CTRL-V-3-predicted single point mutations with experimentally powered Escape Estimator. We extracted the positions from all predicted single point mutations from CTRL-V-3 and leveraged Escape Estimator to gauge the antigenic effects of such arbitrary combinations of mutations. Escape Estimator cross-checking confirms that even though CTRL-V-3 was set up to escape against only one representative antibody (Ly-CoV1404; TM$^×$), these mutations could escape 86% (31 out of 36) of neutralizing antibodies experimentally reported thus far. This again is a significant benchmark in demonstrating that structural priors of DM, TM, and TM$^×$, in concert with the appropriate tapestry of computing steps in CTRL-V-3 truly makes it a propitious, interpretable biosensor design platform as exemplified using the paradigm of viral-escape (as shown by *Escape Estimator*; see S3 Fig).

## Generalization of CTRL-V using the Raf-Ras-Rap1a discriminatory system

To generalize the CTRL-V platform's ability to design a dual bait biosensor, we explored the well-characterized interaction system involving Raf kinase and its differential binding to the GTPases Ras and Rap1a inspired by the study by Serebriiskii *et al* [6] Raf kinase serves as a critical node in the MAPK signaling pathway, where its ability to selectively respond to Ras but not Rap1a (despite their 56% sequence identity) plays a central role in signal transduction specificity. The molecular basis for this discrimination has been extensively characterized. While both Ras and Rap1a interact with Raf's Ras-Binding Domain (RBD) and Cysteine-Rich Domain (CRD), key differences in binding affinity and conformational effects determine signaling outcomes [62,63]. This system represents an ideal test case for generalizing CTRL-V (version 3), as it exemplifies the naturally occurring phenomenon where a protein (Raf) discriminates between structurally similar partners (Ras and Rap1a), preferentially binding to Ras. Using crystal structures of the Ras-Raf complex (PDB: 4G0N) and the Rap1a-Raf complex (PDB: 1C1Y), we applied CTRL-V version 3 to predict mutations in the Raf binding interface, that would further enhance its discrimination between Ras and Rap1a. CTRL-V successfully identified mutations in Raf's RBD that are known [6] to enhance its binding specificity toward Ras. Specifically, interface residue mutations Glu69, Asp66, and Glu84 were corroborated to increase the binding energy differential between Ras and Rap1a interactions (**Fig 8**). The glutamate replaces valine and establishes stronger electrostatic interactions with Ras compared to Rap1a, while the 84E mutation demonstrates how charge distribution affects binding interface complementarity.

## Discussion

We proposed a protein-protein interaction simulation platform where a protein of interest can preferentially discriminate between two proteins by gradually accruing the most favorable mutations over multiple generations. CTRL-V offers a modular platform technology for the mechanistic-design of high-specificity dual bait biosensor proteins. We presented a sequential story for one of the use cases of CTRL-V on three versional implementations of CTRL-V benchmarked against known experimental data on SARS-CoV-2 spike RBD antigen (DM), commercial LY-CoV1404 antibody (TM$^×$), human ACE2 receptor (TM), deep mutational scanning data on the antigen, and 30 circulating variants that emerged from the spike antigen. It is noteworthy that CTRL-V-3 recovers and identifies ~70% (i.e., five out of seven) single point mutations that appeared in the KP.2 variant, which, as of May 2024, is responsible for 28.5% of the infections as per CDC [64]. We have provided a detailed biophysical characterization for KP.2 escape mutations. CTRL-V-3 generalizes to successfully identifying Raf kinase mutations which enable preferential binding with Ras GTPase over Rap1a.

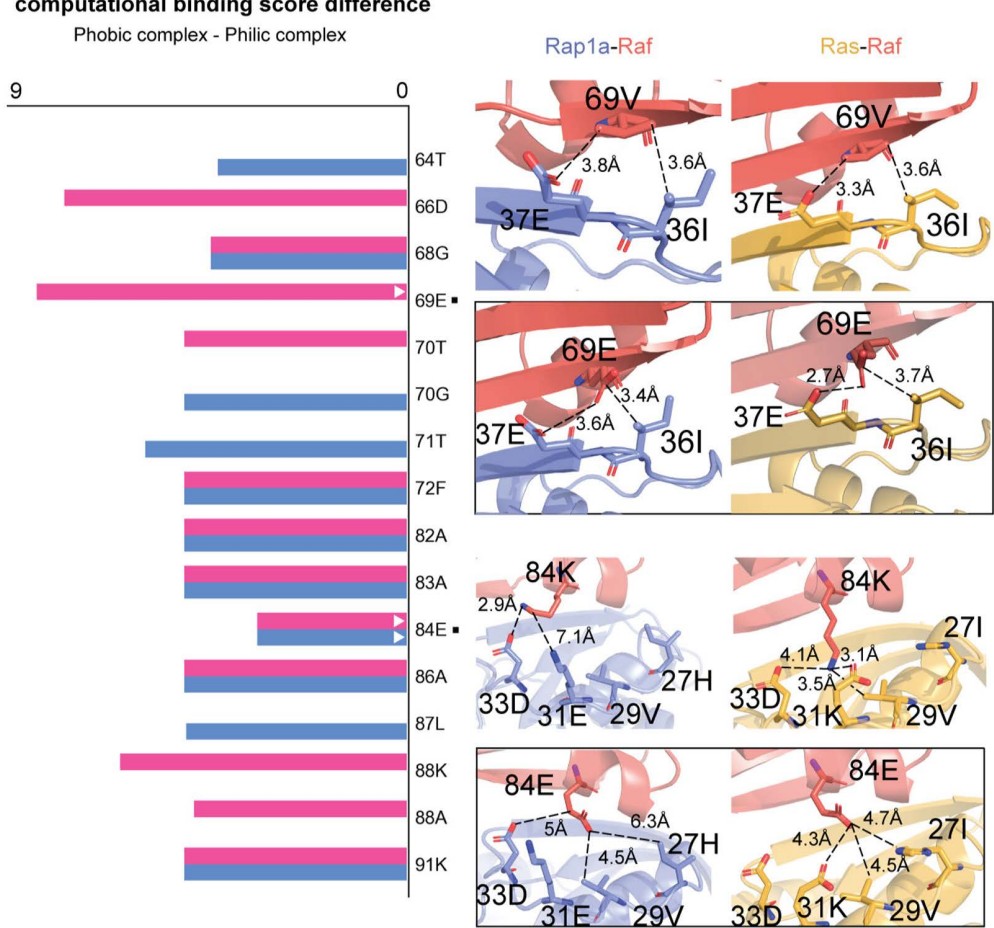

**Fig 8. Computational binding score differences in the Raf-Ras/Rap1a interaction system.** The horizontal bar chart displays differential binding energies (Ras-Raf minus Rap1a-Raf) for key mutations predicted by CTRL-V. Mutations predicted by CTRL-V-3 (mode 1; blue) implementation stabilize Raf's structure while enhancing binding to both Ras and Rap1a, whereas the mode 2 (red) identifies mutations that increase Raf's binding specificity toward Ras at the expense of Rap1a binding, though these may destabilize Raf's overall structure. The right side illustrates a zoomed view of the interface between two proteins with mutated structures placed inside the box below the actual interaction for each highlighted mutation (V69E and K84E). The molecular visualizations highlight two critical mutation sites (V69E and K84E) and their differential interactions with Ras (yellow, right panels) versus Rap1a (blue, left panels). V69E introduces electrostatic interactions at the interface and K84E changes the charged surface to better prefer Ras interface. Distance measurements (in Å) between interacting residues illustrate the structural basis for binding specificity, with closer interactions generally indicating stronger binding.

CTRL-V uses a tilted objective function that maximizes the difference between energy functions, predicting binary variables corresponding to amino acid choices at each position on the polypeptide backbone. This ensures the designed protein iteratively improves upon the current best solution to simultaneously enhance binding to one molecule while eliminating binding with another. The integer optimization model enables a global search of the sequence space with constraints that prevent revisiting previously explored solutions. Although CTRL-V-1, which utilizes integer optimization for sequence design and state-of-the-art tools for structure prediction and docking, yields the worst performance, we rationally discussed the reason for its poor performance. We also provide directions on how it could be improved in future versions and demonstrate it. CTRL-V-1 has an advantage in generating overall global optimal sequences *via* depth-first greedy search, but it is dependent on the amino acid preference score and is vulnerable owing to its

PLOS Computational Biology

dependency on – (a) the low efficacy of the latest structure prediction tools for viral proteins, (b) inability of docking tools to handle constrained multi-chain docking tasks (where only rotamer repacking needs to be done between two chains, and flexible docking needs to be done between the third chain and the common interface of all the three chains). CTRL-V-1 has the lowest recall rate and is GPU intensive due to the ESMfold step. CTRL-V-2 uses a more conventional sequence generation, structure updating, and docking protocols within PyRosetta, which has a moderate recall rate and is the least dependent on multiple tools cross-talking with each other. However, it has a long simulation time and only produces locally optimal sequences as it stochastically explores an umbrella of sequences at each step (breadth first). CTRL-V-3, with the addition of ProteinMPNN, performs best of the three and demonstrates efficacy in pinpointing select point mutations as escape variants. It has the same disadvantages as CTRL-V-2 and is GPU/CPU intensive due to the ProteinMPNN.

CTRL-V is an open-source tool that integrates multiple off-the-shelf computational biology tools (ESMfold, HAD-DOCK3, PyRosetta, ProteinMPNN) with our novel integer optimization code in a single iterative framework. Users need to upload protein structure files (PDB format) for the paired protein complexes (antigen-antibody and antigen-receptor), with the final output delivered as a ranked list of point mutations in CSV format containing binding energy differences and structural metrics. The implementation is available through Google Colab for accessibility across computing environments. Our study was performed on modest hardware specifications (1 node with 8 processor cores, 64GB memory, NVIDIA A100 GPU), where we observed a non-linear increase in true-positive recovery rate with increasing generations of escape variants (**Fig 9**). A more improved CTRL-V with a longer duration can potentially recover more single-point mutations that appear for a given antigen (DM)-antibody (TM*)-receptor (TM) biosensor triad (**Fig 9**; see S3 File). The framework's modular architecture distributes computational load across three key steps, each can be optimized for specific hardware resources. PyRosetta's structure

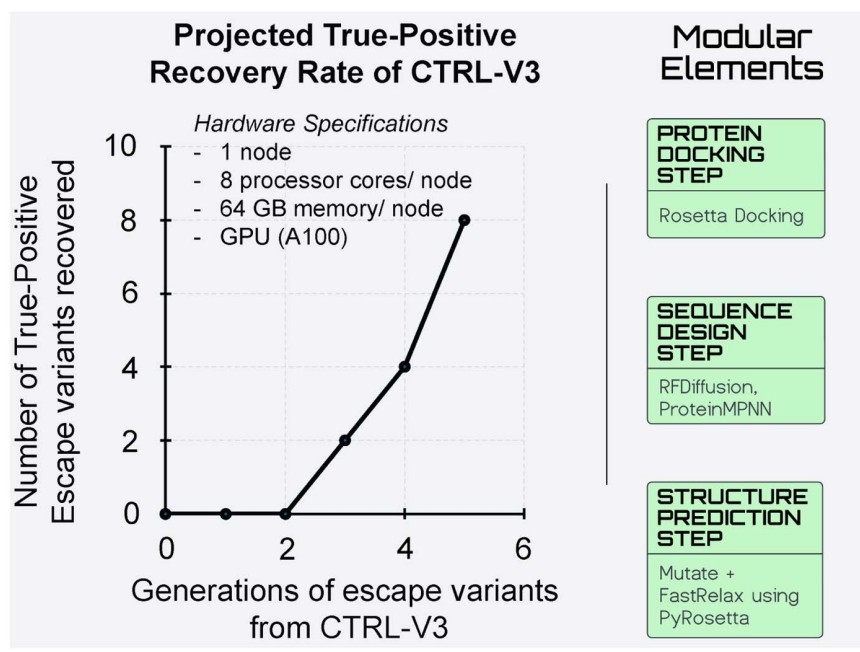

**Fig 9. Hardware specifications used in the SARS-CoV-2 benchmarking study.** When run for longer, i.e., more generations of variants, CTRL–V version 3 can potentially capture more true positives escape mutations, albeit at the cost of higher false positives. Green boxes refer to the choice of modular elements of CTRL–V for this hardware benchmarking task.

refinement employs gradient-based minimization that benefits from CPU thread-level parallelism. The sequence design step using ProteinMPNN can be optimized through GPU acceleration which demonstrated near-linear scaling on GPU clusters [65]. These specifications represent minimum requirements for effective implementation, with performance expected to improve on more powerful computing infrastructure. ProteinMPNN empowers the model to make structure-informed decisions regarding sequences that conform to the antigenic viral protein (DM) structure, yielding robust coverage of high-fidelity sequences that likely fold into the same structure. Following this proof-of-concept analysis of the results, we recommend utilizing the preliminary CTRL-V-3, as a promising viral escape-inspired, dual bait biosensor design platform. This version is accessible online through Google Colab freely under Creative Commons License CC-3.0 (https://github.com/ChowdhuryRatul/CTRL-V).

## Methods

### Choice of tools that were streamlined and leveraged for benchmarking of CTRL-V

In developing CTRL-V, we implement state-of-the-art software tools in bioinformatics. ESMfold (Evolutionary Scale Modeling) is a deep learning-based method developed to predict protein structure from a sequence. This allows us to update the structure of the protein during each mutation [66]. Haddock3 (High Ambiguity Driven DOCKing) is an information-driven approach for modeling biomolecular complexes. Using two disembodied proteins as input, Haddock3 returns a complex protein that tells us how two of the proteins interact structurally. This allows us to obtain interaction poses between antibodies and viral proteins [67,68]. PyRosetta is a protein modeling software that includes algorithms for computational modeling and analysis of protein structures. Using PyRosetta, we perform single point mutation to the protein and obtain the binding energy information between two proteins at specific binding poses [69]. ProteinMPNN is the latest deep learning-based method developed for protein sequence design. From a given protein structure, ProteinMPNN tells us what other sequences can likely fold into the same structure [67]. This allows us to narrow down the possible amino acid mutational landscape for the viral protein. These four software tools are independent on their own and have been integrated and automated to be executed during each design cycle of CTRL-V. While these tools have been explored within CTRL-V, it remains purely modular in its construction, permitting the user to leverage any other tool for the relevant steps, including – (a) sequence design, (b) structure prediction, (c) docking, (d) sequence evaluation through energetics, and (e) acceptance and rejection criterion of a design.

### Integer optimization model for CTRL-V-version 1

We developed an integer linear optimization programming framework that solves the Favorable Mutation Problem (FMP) and generates a library of viral escape sequences (workflow; **Fig 2**). Integer optimization for sequence design through structure-informed modeling has been demonstrated in prior work to hold experimental fidelity in designing enzymes for altered substrate [39,70,71] and cofactor [72] specificity, protein pores [73] for altered pore size, and affinity maturation [74] of *de novo* designed antibodies [75]. CTRL-V-1 implements such an integer optimization model towards the FMP, wherein the wild type of SARS-CoV-2 spike protein (antigen) sequence is encoded in integer format, scored for mutations that enable viral escape, and returns the top-*k* preferred mutations as shown in the pseudocode built with Pyomo [76,77].

Jha *et al.* tabulated the interaction preference score for all 400 pairs of 20 amino acids [54], where a higher number in the table indicates a stronger repulsive force between the affiliated amino acid pairs. The integer optimization model utilizes this preference score for sequence design for both the viral as well as the antibody protein, i.e., identifying point mutations that allude to the objective of improving binding to the receptor and simultaneously alleviating binding with the antibody (also referred to as the Favorable Mutation Problem - FPM). The distance between amino acids in the viral protein and antibody varies depending on the geometry of the interface.

**Algorithm 1 – Multi optima search (pseudo code)**

_Step 1._ optimal_solutions = new set().

_Step 2._ **while** size(optimal_solutions) ≤ k **do.**

_Step 3._ optimals = OptimizationModel ().

_Step 4._ optimal_solutions.add(optimals[0]).

_Step 5._ remove optimals[0] from feasible solution space.

_Step 6._ **end while.**

_Step 7._ return optimal solutions.

The integer linear programming formulation has been explained as follows. Here, we define the variable _m_ (equivalently, _n_ and _p_) as amino acid type, _i_, _j_, and _k_ are amino acid positions on the antibody, antigen, and receptor sequences, respectively.

**Sets**

$M \equiv N = \{m, n | Ala, \ldots, Tyr\}$: list of all 20 amino acids.
$I = \{i | 1, \ldots, L_{TM^\times}\}$: positions on the antibody ($TM^\times$) sequence.
$J = \{j | 1, \ldots, L_{DM}\}$: positions on the antigen (DM) sequence.
$K = \{k | 1, \ldots, L_{TM}\}$: positions on the receptor (TM) sequence.

**Parameters**

$C_{mn}$ = preference score of the $m^{th}$ and $n^{th}$ amino acid pairs.
$d_{ij}$ = distance between antibody amino acid in the $i^{th}$ position and antigen amino acid in the $j^{th}$ position.
$E_{im}^{jn}$ = Scaled preference score of amino acid type _m_ in the $i^{th}$ position on Ab and amino acid type _n_ in the $j^{th}$ position on Ag, where they are separated by $d_{ij}$ distance (is given by)

$$E_{im}^{jn} = \frac{C_{mn}}{d_{ij}^2}$$

(1)

We convert protein sequences into integer format, using a binary parameter.

$$x_{im} = \begin{cases} 1 & \textit{if the ith } TM^\times \textit{ position is occupied by amino acid type m)} \\ 0 & \textit{otherwise} \end{cases}$$

(2)

$$z_{kp} = \begin{cases} 1 & \textit{if the kth TM position is occupied by amino acid type p} \\ 0 & \textit{otherwise} \end{cases}$$

(3)

**Variables**

Similarly, we represent evolving antigen sequence (design molecule) using a binary variable

$$y_{jn} = \begin{cases} 1 & \textit{if the } j^{th} \textit{ DM position is mutated to amino acid type n} \\ 0 & \textit{otherwise} \end{cases}$$

(4)

## Integer optimization formulation for dual bait biosensing

The design objective within the integer optimization model is to identify amino acid transitions at different residue positions on the antigen (design molecule; DM), to improve binding to the receptor (target molecule; TM) and minimize binding with the antibody (non-target molecule; $TM^{\times}$). While this should be ideally cast as a bilevel objective where the minimization problem is nested within an outer maximization problem, we will use a tilted objective function where we maximize the difference between antigen binding with receptor and antibody. Since the $C_{mn}$ scores are negative for favorable interactions, and positive for unfavorable interactions, minimizing the objective function leads to identifying amino acid transitions that stabilize the design molecule and increase the interaction gap between DM-TM and DM-$TM^{\times}$.

$$Maximize \sum_{i \in I} \sum_{m \in M} \sum_{j \in J} \sum_{n \in N} E_{i,m}^{j,n} x_{im} y_{jn} \sum_{k \in K} \sum_{p \in P} \sum_{j \in J} \sum_{n \in N} E_{i,m}^{j,n} z_{kp} y_{jn}$$

such that,

$$\sum_{n=1}^{20} y_{jn} = 1 \quad \forall j \in \{1, ..L_{DM}\} \tag{5}$$

$$\sum_{j=1}^{L_{DM}} \sum_{n=1}^{20} y_{jn} = L_{DM} \tag{6}$$

$$\{x_{im}, y_{jn}, z_{kp}\} \in \{0,1\} \tag{7}$$

The first term in the objective function calculates all pairwise inter-amino acid interaction scores between residues of the design molecule (antigen, in this case) and the target molecule (receptor), while the second term calculates the between the design molecule and the non-target molecule (antibody). Equation (5) and (6) ensures that exactly one amino acid is chosen at a single position in the DM, and the sum across all positions should equal to the length of the design molecule ($L_{DM}$). Finally, equation (7) is an indicator that x, y, and z are binary parameters and variables, respectively.

This optimization formulation can be easily extended to encode a scenario, where in every next iteration after the DM accrues a mutation to eliminate binding with $TM^{\times}$, the $TM^{\times}$ also finds a mutation to restore the binding with DM. This is evocative of the process of antibody affinity maturation for viral infection. This makes this a moving target problem, where DM engages in an iterative tug-of-war to escape from a dynamically adversarial $TM^{\times}$ sequence. This is also important in the paradigm of a dual bait biosensor where some mutation to the DM makes it non-competitively inhibited by $TM^{\times}$ leading to loss of binding with TM. Capturing the nuances of point-mutation-induced drastic structural change is beyond the scope of CTRL-V, but equations (8) and (9) could be added to the workflow to solve the moving target ($TM^{\times}$ sequence changing) problem.

$$\sum_{m=1}^{20} x_{im} = 1 \quad \forall i \in \{1, ..L_{TM^{\times}}\} \tag{8}$$

$$\sum_{i=1}^{L_{TM^{\times}}} \sum_{n=1}^{20} x_{im} = L_{TM^{\times}} \tag{9}$$

It is to be noted that, for the moving target problem, $x_{im}$ is a variable and not a parameter.

CTRL-V-1, therefore, implements a greedy best-first search (GBFS) technique that simulates the process of viral escape. GBFS prioritizes the choice that appears to be the most promising and builds rounds of mutations each progressively improving on the objective function. We simulate the viral escape process by iteratively selecting mutations on DM (antigen) that create highest discrimination between the antibody and receptor proteins. We include critical observations and maintain parity with the true nature of viral mutation, including (a) high concentration of mutations at the RBD region for immune escape [11,57], (b) only rare instances of mutation at the same polypeptide locus on the viral spike protein in consecutive generations [11], (c) most mutations being neutral or deleterious towards the FMP, and (d) most stable mutations cause least impact on the viral protein structure [56].

### CTRL-V version 1: Integer programming approach *for* CTRL-V-1

In CTRL-V-1, we address the Favorable Mutation Problem through the integer programming to predict the mutations to RBD (Equations 1–9). This workflow (**Fig 2**) requires an antigen-antibody complex structure as input, wherefrom the inter-residue distance information at the protein-protein interface is obtained and converted into an integer representation. Using the amino acid pairwise interaction score (as described above), new sequences for the antigen (and optionally, for the antibody, reflecting simultaneous antigen escape and antibody affinity maturation) are predicted per iteration in the integer representation. After recovering the new mutant antigen (DM) sequence, the corresponding protein structure is predicted using ESMfold. The new antigen structures are re-docked on the antibody (and receptor) using HADDOCK3 to get the new interaction interface geometry (inter-residue distance matrix) – specifically, the amino acid distance at the DM-TM (and DM-$TM^x$) interface(s). This workflow runs iteratively in loops and, returns a list of predicted single-point mutations without revisiting the same solution (integer cuts). CTRL-V-1, thus, simulates the viral protein escaping (i.e., biosensor evolving) from the antibody. It is noteworthy that CTRL-V is also, by design, equipped to handle a bidirectional complex *tug-of-war* between protein pairs (antigen: DM and antibody: $TM^x$), where both DM and $TM^x$ are allowed iteratively mutate. The antigen (DM) mutates to lower, and the antibody ($TM^x$) mutates to restore/ enhance the binding affinity between them. This represents a unique execution of an integer optimization with an iteratively moving target. At each step the antigen-receptor (DM-TM) binding evaluations dictate the best DM variant. The best variant will have the highest energy gap between DM-TM and DM-$TM^x$ complexes.

### CTRL-V version 2: PyRosetta sequence design and rigid docking approach *for* CTRL-V-2

While an integer programming approach guarantees a global optimum and no stochasticity across runs, it is a depth-first (greedy) search where mutations are progressively built on one best-performing mutant – it leaves room for breadth search (i.e., exploring multiple different sequence design trajectories). To this end, in CTRL-V-2, we used PyRosetta (**Fig 2**) to perform single-point mutation to the viral protein stochastically. Also, in accordance with Sanjuán *et al* [56]. we reflect minimal changes to the viral backbone structure upon mutation, ensuring side chain repacking. Much like the first version, CTRL-V-2 also takes antigen-antibody complex and antigen-receptor complex structure files as inputs. CTRL-V-2 first identifies interacting amino acids in the complex, likely to be epitopes of the viral spike protein. It passes the list of epitope positions and the two complex files to PyRosetta. PyRosetta then ranks all possible 20-point mutations in all these positions that lower binding with antibody without compromising binding with receptor (i.e., FMP). The top-ranking mutations against both antibody and receptor are selected and used to update the antigen protein structures in both complexes using PyRosetta. After the update, the workflow automatically identifies the supplemental amino acid choices in subsequent iterations. This workflow runs iteratively in a loop along multiple design trajectories and, at the end, returns a list of predicted single-point mutations. CTRL-V version 2 simulates the viral protein escaping from the antibody, ensuring that the mutated virus is favorable for entry into human cells.

**CTRL-V version 3: Protein MPNN-based sequence design and rigid docking *in* CTRL-V-3**

Herein, we extend the ability of CTRL-V to leverage ML-based sequence design using the deep learning-based model – ProteinMPNN (**Fig 2**). Compared to CTRL-V-2, this version takes antigen-antibody complex and antigen-receptor complex files and first generates, using ProteinMPNN, an atlas of sequence predictions for the antigen as input. CTRL-V-3 uses the list of sequences by ProteinMPNN as the potential mutations for the viral spike protein (DM), which do not disrupt the secondary structure of the spike RBD (antigen) and only ranks these potential mutations against the receptor (TM) and antibody (TM$^{\times}$). The top-ranking mutations against both antibody and receptor are selected and used to update both complexes using PyRosetta. After the update, the system again queries the list of permitted sequences from ProteinMPNN for subsequent design iterators. Contrary to CTRL-V-2, the deep learning model enables the exploration of combinatoric point mutations at different antigen loci simultaneously, albeit at a higher computational cost. ProteinMPNN also allows the generation of the sequence atlas using different biophysical constraints, such as - permitting the user to define any arbitrarily sized design window on the antigen protein only where mutations will be permitted while keeping other parts of the antigen unaltered. This workflow runs iteratively in a loop and, at the end, returns a library of predicted single-point mutations that satisfy the FMP. CTRL-V-3, with ProteinMPNN, simulates the antigen (DM) protein sequences that (a) maintain the antigen's secondary structure and (b) enable the virus to escape from the antibody while retaining favorable entry into the human host. CTRL-V's main utility is for a dual bait biosensor design platform, honed over three model architecture iterations with viral escape data to assess model quality, and shown to generalize in non-viral systems (Raf kinase signaling), rather than aiming to be an escape mutation predictor.

### Significance statement

The ability to predict a clade of likely variants (of pathogens) provides significant biotechnological advantages both nationally and globally, enabling the design of broadly neutralizing therapeutic peptides that remain effective against any future escape variant. This foresight is crucial for maintaining effective countermeasures against (re)emerging pathogenic threats. Additionally, the growing need for biosensors in environmental monitoring, health diagnostics, sustainability, and energy applications underscore the importance of reliable predictive technologies. The receptor binding domain (RBD) of SARS-CoV-2 evolves to evade antibody protection but increase/ maintain receptor binding. Inspired by the structural biology of viral escape, the CTRL-V platform is optimized to design protein-based dual bait biosensors that discriminate between two proteins and can be extended towards the selective binding between binary pairs of a wide portfolio of macromolecules (metal ions, small molecules, peptides, and combinations thereof).

### Supporting information

**S1 Fig. Pairwise amino acid interaction scores empirically measured by *Jha et al.* [45].** A more negative score represents strong attractive forces between two amino acids separated by optimal bonding distance (from Lennard-Jones potential; LJ 6–12 equation). The table is however biased away from Proline and towards Cysteine, i.e., pairwise preference scores for any amino acid with Pro are very high (positive) while those with Cys are very low (negative). This creates an imbalance in the integer optimization protocol introduced in CTRL-V-1 (see *Methods*).
(TIFF)

**S2 Fig. Visual representation of choice of different amino acids for viral escape at the interface of SARS-CoV-2 spike protein and Ly-CoV1404 antibody across multiple viral escape trajectories predicted by CTRL-V-1.** The bright vertical swatches represent the interacting residues from the spike protein (x-axis represents all amino acids on the viral spike RBD domain from N-terminus through C-terminus). The two horizontal streak likes - one bright (for Pro) and one dark (for Cys) indicates the super high and low preference for Pro and Cys during mutations on the spike protein. The choice is biased due to the imbalance in the pairwise amino acid preference score (S1 Fig).
(TIFF)

**S3 Fig. Escape Estimator on CTRL-V-3 variants reveal that 86% of the sites identified as potentially escape are truly loci for accruing escape mutations.** While CTRL-V-3 was only provided information about the Ly-CoV1404 neutralizing antibody to predict escape loci, the predictions recovered loci which, through mutations, aids the SARS-CoV-2 virus to escape from 36 reported neutralizing commercial antibodies.
(TIFF)

**S1 File. List of 30 most popular SARS-CoV-2 variants circulating globally over the past three years as reported by Covariants, containing 39 different single-point mutations used for benchmarking CTRL-V.**
(XLSX)

**S2 File. Experimental $\log_{10}$ binding and expression scores from deep mutational scanning of the SARS-CoV-2 receptor binding domain by Greaney et al.[58], used for comparison with CTRL-V predictions.**
(XLSX)

**S3 File. Recovery rate of true-positive mutations with increasing generations of escape variants, demonstrating the potential of CTRL-V for predicting viral escape mutations with longer simulation durations.**
(XLSX)

## Author contributions

**Conceptualization:** Ratul Chowdhury.

**Data curation:** Yee Chuen Teoh, Mohammed Sakib Noor, Jack Girton.

**Formal analysis:** Yee Chuen Teoh, Mohammed Sakib Noor, Guiping Hu.

**Funding acquisition:** Ratul Chowdhury.

**Investigation:** Yee Chuen Teoh, Ratul Chowdhury.

**Methodology:** Yee Chuen Teoh, Mohammed Sakib Noor, Sina Aghakhani, Ratul Chowdhury.

**Project administration:** Ratul Chowdhury.

**Resources:** Ratul Chowdhury.

**Supervision:** Ratul Chowdhury.

**Visualization:** Mohammed Sakib Noor, Ratul Chowdhury.

**Writing – original draft:** Yee Chuen Teoh, Ratul Chowdhury.

**Writing – review & editing:** Mohammed Sakib Noor, Sina Aghakhani, Jack Girton, Guiping Hu, Ratul Chowdhury.

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
