## [Decision Letter · Decision Letter 0]

23 Jan 2025

PCOMPBIOL-D-24-01488

Viral Escape-Inspired Framework for Precision Structure-guided Protein Biosensor Development

PLOS Computational Biology

Dear Dr. Chowdhury,

Thank you for submitting your manuscript to PLOS Computational Biology. After careful consideration, we feel that it has merit but does not fully meet PLOS Computational Biology's publication criteria as it currently stands. Therefore, we invite you to submit a revised version of the manuscript that addresses the points raised during the review process.

Please submit your revised manuscript within 60 days Mar 25 2025 11:59PM. If you will need more time than this to complete your revisions, please reply to this message or contact the journal office at ploscompbiol@plos.org. Please include the following items when submitting your revised manuscript:

We look forward to receiving your revised manuscript.

Kind regards,

Claudio José Struchiner, M.D., Sc.D.

Academic Editor

PLOS Computational Biology

Arne Elofsson

Section Editor

PLOS Computational Biology

**Additional Editor Comments:**

The review process highlights the necessity for significant revisions in the methodology description, validation, and presentation to enhance clarity, reproducibility, and scientific rigor.

**Journal Requirements:**

5) Please ensure that the funders and grant numbers match between the Financial Disclosure field and the Funding Information tab in your submission form. Note that the funders must be provided in the same order in both places as well.

**Reviewers' comments:**

Reviewer's Responses to Questions

**Comments to the Authors:**

Reviewer #1: Teoh et al have presented a manuscript that describes the mechanistic design of a high specificity biosensor protein. They describe use cases of 3 versions of differing versions of the program (CTRL-V) benchmarked against known experimental data on SARS-CoV-2 spike antigen, a monoclonal antibody, a human receptor and data from a DMS experiment and 30 circulating variants emerging from Spike. The manuscript covers the performance of the 3 iterations of the model in predicting true variants compared to false variants. CTRL-V-3 performs the best amongst the 3 when compared to these experimentally derived datasets and is beneficial in that it requires no training phase. I believe that this study is suitable for publication with the caveat, that this is not my exact field of expertise. The experimental design appears to be sound, although parts of the paper the relevance of the method should be clarified.

Reviewer #2: The paper shows a potential method that can be used to make a biosensor. However, I have following concerns and suggestions:

1. Is it safe to use the COVID-19 virus to construct a bio sensor (we hate the virus). Please justify this choice and state the measurement for guarantee the safety for the public.

2. Why the specific commercial LY-CoV1404 antibody is used, will other brand or other antibodies also fit in the method along with the framework?

3. Why only hydrophobicity is considered important here? Are there any other indexes that is also important?

4. The presentation of the entire paper needs a major modification, please separate the discussion on biological termed analysis (for biologist) with the method explanation used like integer optimization formulation (for engineer), and simulation procedures (for programmer) as well.

5. It looks the entire paper is comparing pro and con for the gradually improved platforms, from which, the reader can neither gain inside of the virus mutation escaping antibody, nor the inside of a potential flexible sensor design, or the appreciation of the accuracy of the Python model.

5. Please explain the importance of electrostatic contact, how it is compared with hydrophobicity?

6. Please number all the formulae.

7. Why there are a number of 0% in the simulation table, is it caused by the particular virus sequence or computer code/model design?

Reviewer #3: Recommendations to Authors

Highlight Novelty: Emphasize how CTRL-V differs from existing escape-prediction and biosensor design frameworks. Include quantitative comparisons to establish its advantage.

Enhance Methodological Clarity: Provide detailed descriptions of integer programming, PyRosetta, and ProteinMPNN steps. Ensure all methods are reproducible by detailing parameter choices and workflow configurations.

Improve Validation:

Strengthen computational validation by comparing CTRL-V outputs with experimental datasets from other viral escape studies or biosensor applications.

Consider including wet-lab validation to support computational predictions.

Focus on Presentation:

Use clearer, high-resolution images for structural results and create more descriptive legends.

Reorganize tables to facilitate comparisons across CTRL-V versions.

Expand Generalizability: Provide examples or mock results to demonstrate CTRL-V's potential applications beyond SARS-CoV-2.

Address Scalability: Discuss computational resource requirements and how CTRL-V can be scaled or optimized for broader adoption.

**Have the authors made all data and (if applicable) computational code underlying the findings in their manuscript fully available?**

Reviewer #1: Yes

Reviewer #2: Yes

Reviewer #3: Yes

PLOS authors have the option to publish the peer review history of their article (what does this mean? ). If published, this will include your full peer review and any attached files.

**Do you want your identity to be public for this peer review?** For information about this choice, including consent withdrawal, please see our Privacy Policy .

Reviewer #1: No

Reviewer #2: **Yes: ** Jun Steed Huang

Reviewer #3: No

**Figure resubmission:**
---

## [Decision Letter · Decision Letter 1]

14 Mar 2025

Dear Mr Chowdhury,

We are pleased to inform you that your manuscript 'Viral Escape-Inspired Framework for Precision Structure-guided Dual Bait Protein Biosensor Development' has been provisionally accepted for publication in PLOS Computational Biology.

Best regards,

Claudio José Struchiner, M.D., Sc.D.

Academic Editor

PLOS Computational Biology

Arne Elofsson

Section Editor

PLOS Computational Biology

Reviewer's Responses to Questions

**Comments to the Authors:**

Reviewer #2: Good revision.

**Have the authors made all data and (if applicable) computational code underlying the findings in their manuscript fully available?**

Reviewer #2: Yes

PLOS authors have the option to publish the peer review history of their article (what does this mean? ). If published, this will include your full peer review and any attached files.

**Do you want your identity to be public for this peer review?** For information about this choice, including consent withdrawal, please see our Privacy Policy .

Reviewer #2: **Yes: ** Jun Steed Huang

---

## [Editor Report · Acceptance letter]

PCOMPBIOL-D-24-01488R1

Viral Escape-Inspired Framework for Structure-guided Dual Bait Protein Biosensor Design

Dear Dr Chowdhury,

I am pleased to inform you that your manuscript has been formally accepted for publication in PLOS Computational Biology. Your manuscript is now with our production department and you will be notified of the publication date in due course.

With kind regards,

Anita Estes
